# Prospective, Cross-Sectional Study Finds No Common Viruses in Cerebrospinal Fluid of Children with Pseudotumor Cerebri

**DOI:** 10.3390/brainsci13020361

**Published:** 2023-02-19

**Authors:** Rony Cohen, Muhammad Mahajnah, Yulia Shlonsky, Orit Golan-Shany, Azriel Romem, Ayelet Halevy, Keren Natan, Jacob Genizi

**Affiliations:** 1Department of Pediatric Neurology, Schneider Children’s Medical Center of Israel, Petah Tikva 4920235, Israel; 2Sackler Faculty of Medicine, Tel Aviv University, Tel Aviv 69978, Israel; 3Pediatric Neurology Unit, Hillel Yaffe Medical Center, Hadera 38100, Israel; 4Bruce Rappaport Faulty of Medicine, Technion (Israel Institute of Technology), Haifa 3104802, Israel; 5Microbiology Labratory, Bnai Zion Medical Center, Haifa 3104802, Israel; 6Pediatric Department, Bnai Zion Medical Center, Haifa 3104802, Israel

**Keywords:** pseudotumor cerebri, viruses in CSF, etiology

## Abstract

Pseudotumor cerebri (PTC) in children is a rare condition whose underlying cause remains largely unknown. No study has yet systematically examined viral infection as a cause of PTC. The current study aimed to characterize PTC in children and investigate the possible role of acute viral infection of the central nervous system in its pathogenesis. A prospective, cross-sectional study was conducted in three centers in Israel. Participants were 50 children aged 0.5–18 years, of whom 27 had a definitive diagnosis of pseudotumor cerebri (the study group) and 23 comprised a control. Data collected included clinical presentation, imaging, treatment, ophthalmic findings, and cerebrospinal fluid (CSF) analysis. Using the ALLPLEX^TM^ meningitis panel, real-time polymerase chain reaction (PCR) was used to test for the presence of 12 common viruses. PTC patients (mean age 12 ± 4.3 years; 14 males, 13 females) had mean opening pressure of 41.9 ±10.2 mmH2O. All PTC patients had papilledema, and 25 (93%) had PTC symptoms. No viruses were found in the PTC group, while in the control group, one patient tested positive for Epstein–Barr virus and another for human herpesvirus type 6. Overall, in our study, PTC was not found to be associated with the presence of viruses in CSF.

## 1. Introduction

Pseudotumor cerebri (PTC), also known as idiopathic intracranial hypertension (IIH), is an uncommon condition in children, with a reported incidence of 0.60–0.71 per 100,000 children. The revised Freidman criteria [1] for definite pseudotumor cerebri syndrome are as follows: elevated opening cerebrospinal fluid (CSF) pressure (above 280 mm H_2_O in sedated children); papilledema or abducens nerve palsy; and normal CSF composition without an identified cause for increased intracranial pressure (ICP) in neuroimaging. Pseudotumor cerebri usually presents with such symptoms as headache, nausea, vomiting, blurred vision, and irritability [2]; however, it can also be asymptomatic. Without treatment, pseudotumor cerebri may have serious effects on vision, potentially leading to optic atrophy and decreased visual acuity. The underlying cause of pseudotumor cerebri remains largely unknown. Predisposing factors include various medications, among them tetracycline, corticosteroids, and nalidixic acid; vitamin A supplementation; anemias; endocrinopathies; and being overweight. 

Infections, such as meningitis and encephalitis, are known causes of intracranial pressure. However, the basic criterion of normal CSF content [1] should exclude such etiologies for PTC. Tamer et al. [3] described cases of PTC related to viral illness; however, they did not show a direct connection. 

There is a relative paucity of literature on viral central nervous system infection as a potential cause of increased ICP in PTC. It should be emphasized that the absence of CSF pleocytosis does not rule out CNS infection. Only case reports or small case series have been described, relating variously to the hepatitis E virus [4], hepatitis A virus [5], measles [6], varicella [7,8,9,10], enterovirus [11,12,13], and recently, SARS-CoV-2 as well [14,15,16]. However, no study has systematically used polymerase chain reaction (PCR) to test for viral infection in the CSF of pediatric patients with PTC.

The aim of the current study was to characterize PTC in children and to investigate the possible role of acute viral infections of the central nervous system in the pathogenesis of pseudotumor cerebri.

## 2. Materials and Methods

The present prospective, cross-sectional study was conducted in three centers in Israel: the Bnai Zion, Hillel Yaffe, and Schneider Children’s Medical Centers. Twenty-seven children aged 0.5–18 years with pseudotumor cerebri were prospectively studied. Ethical approval was obtained from each hospital, and all participants or caregivers provided written informed consent. All children were examined by a pediatric neurologist and ophthalmologist. The data collected included clinical presentation, imaging, treatment, ophthalmic findings, and CSF analysis. The CSF was taken as part of the PTC diagnosis at the time of measuring the opening pressure. Using the ALLPLEX^TM^ meningitis panel, real-time PCR was used to test for the presence of 12 viruses that are known to cause infections of the central nervous system: cytomegalovirus, Epstein–Barr virus, herpes simplex virus 1 and 2, human herpesvirus 6 and 7, varicella–zoster virus, adenovirus, parvovirus B19, human parechovirus, measles, and hepatitis E (https://www.seegene.com/assays/allplex_meningitis_panel_assays , accessed on 1 December 2022). Twenty-three pediatric CSF samples without PTC or known infection were also tested using the ALLPLEX^TM^ meningitis panel and served as controls. The samples were kept at −80 Celsius, and the PCR panel was performed at the end of the study. All precipitants were vaccinated against varicella–zoster virus but not against any other pathogen included in the study.

## 3. Results

We collected and analyzed CSF samples from 50 children, of whom 27 had a definitive diagnosis of pseudotumor cerebri, and 23 served as a control group.

Table 1 presents the characteristics of the two groups. The mean age of the patients with pseudotumor cerebri was 12 ± 4.3 years; 14 were males and 13 were females. Nine of the 27 PTC patients were under 12 years of age, including six males and three females (a male-to-female ratio of 2:1). All the patients had papilledema. The mean opening pressure was 41.9 ±10.2 mmH2O. Twenty-five children (93%) had symptoms upon presentation, while 2 (7%) were asymptomatic; in those children, PTC was discovered incidentally during an ophthalmology evaluation. The presenting symptoms were headache (19 patients, 70%), blurred vision (14 patients, 52%), tinnitus (6 patients, 22%), and nausea and vomiting (6 patients, 22%), see Table 2. 

No viral infections were found in the PTC group. In the control group, one patient tested positive for Epstein–Barr virus and another for human herpesvirus type 6.

## 4. Discussion

In our study, we looked at whether an acute CNS infection can be a cause of PTC in children. We studied the CSF of children with PTC and compared them to the CSF of a control group. In our study, we performed the PCR of 12 common CNS viral pathogens and found no pathogens in the PTC group. It must be emphasized that our study, although conducted in three different medical centers, was conducted in Israel, and other regions or populations may have different common viral pathogens that we have not sought out.

The pathogenesis of pseudotumor cerebri is still unknown [17]. Although transverse or sigmoid sinus stenosis or atrophy are frequently seen in children with PTC [18], this is probably not the primary cause but secondary to the increased pressure [17]. However, stenosis may worsen intracranial pressure by interfering with CSF removal. The association of PTC with female gender, obesity, and polycystic ovary syndrome may suggest that aldosterone could play an important factor by increasing the activity of Na/K-ATPase in the choroid plexus; however, this theory is not supported due to the absence of morphological changes in the choroid plexus. The high prevalence of obesity and the occurrence of PTC after administration of certain medications [19] might be explained by the effects of estrogen or retinoic acid on epithelial cells, leading to less CSF outflow. Genetic studies on AQP4, which facilitates water flow into and out of the brain, also found no association [17,20]. Other possibilities mentioned are low-grade inflammation and dysfunction in the glymphatic pathway [21].

Although both Dandy’s original criteria [22] and the revised criteria [1] exclude CSF pleocytosis, these criteria do not necessarily exclude infection. Before new molecular technologies such as PCR were available, the absence of pleocytosis presumably excluded meningeal infection. However, we now know that viral meningitis may not always be associated with pleocytosis [23,24]. Parecho virus [25], enterovirus [26], and even herpes simplex virus type 1 [27] may present with no pleocytosis. Since meningeal infection can cause intracranial pressure, this raises the possibility that viral meningitis without pleocytosis might be the cause of PTC in some patients. 

Viral infection of the CNS increases the levels of inflammatory cells such as chemoattractants, neutrophils, CD8 T cells, and monocytes, suggesting the stimulation of the immune system. The immune response is mediated by cytokines such as IL-1β IL-6 and INF-γ in CSF [28]. This inflammatory process may be the cause of elevated intracranial pressure in meningitis. 

Some case reports have connected PTC to viral infections. Tamer et al. [3] report fifteen cases of PTC in infants related to viral infection either before or during the illness. All infants showed complete improvement within a few weeks, and only a third had to be treated with acetazolamide. Tamer et al. coined the term “benign infantile viral PTC” based on these patients. However, this group is unique. The CSF opening pressure is not mentioned, and no other reports have been made since. 

Konrad et al. [7] reported a case of PTC three weeks after varicella infection. However, they did not measure the CSF opening pressure, and although this patient was proven to have iliofemoral thrombosis, CT excluded intracranial thrombosis. Kan et al. [28] gathered twelve cases from the literature of PTC associated with Lyme disease. However, pleocytosis was documented, making these cases of encephalitis with intracranial pressure and not PTC according to Dandy’s criteria [1]. Taşdemir et al. [6] reported a case of PTC following measles infection, and Thapa and colleagues [4,5] reported two cases of PTC following hepatitis A and E infection. However, in both cases, the association between the infection and PTC was not proved (Table 3).

Ravid et al. [8] described three children with PTC as a manifestation of varicella–zoster virus reactivation. However, only one of their patients did not have pleocytosis and thus met the criteria for PTC. Another four patients with PTC following varicella–zoster virus infection have been reported in the literature [9,10]. However, two did not have high opening pressure, and a third had pleocytosis. Given reports [13] that enteroviral meningitis may cause increased intracranial pressure, we, therefore, agree with the contention of Ali et al. [10] that varicella–zoster meningitis may be a mimicker of PTC rather than its origin. In our own experience, we also saw two children with enteroviral meningitis and increased intracranial pressure who did not meet the criteria of PTC. We can conclude that, to date, most published papers connecting PTC to viral infection have not proved the connection. 

The only papers that reported PTC during acute illness are Millichap [9] and McMinn [11]. Millichap [9] reported on a 14-year-old girl with acute herpes zoster, and McMinn [11] reported on a 6-year-old girl with Enterovirus 71. Both had CSF pleocytosis (Table 3).

Our study systematically examined the cerebrospinal fluid of a group of children with PTC using PCR to test for common viruses. In our study, no viruses were found in the CSF of children diagnosed with PTC. Our findings thus support the initial assumption that PTC is not caused directly by a viral pathogen.

Limitations: Our paper has some limitations. The sample is small. We studied only CSF and not blood samples. Hepatitis A and enterovirus that previously have been connected to PTC were not part of our panel. Although PCR is a good choice for acute infection, it does rule out a post-infection process.

## 5. Conclusions

In conclusion, in our study, based on real-time PCR, pseudotumor cerebri was not found to be associated with the presence of viruses in CSF, as compared to a control group. Our study has two main limitations. First, we examined a specific, although common, panel of viruses. Second, we examined CSF for the presence of viruses after the patient was diagnosed with PTC. Thus, our findings do not rule out the possibility that PTC was caused by a prior infection where the pathogen was no longer present in the CSF.

## Figures and Tables

**Table 1 brainsci-13-00361-t001:** Characteristics of the PTC and control groups.

	Girls	Boys	Total	*p*	Age (Years)	*p*
PTC	13 (48%)	14 (52%)	27	*p* = 0.35	12.1 (4.3–17.7)	*p* = 0.29
Control group	7 (30%)	16 (70%)	23		10.7 (0.05–17.4)	

**Table 2 brainsci-13-00361-t002:** Clinical presenting symptoms of the PTC and control groups.

	PTC	Control
Headache	19 (70%)	8 (35%)
Vomiting	6 (22%)	4 (17%)
Visual changes	14 (52%)	7 (30%)
Tinnitus	6 (22%)	0
Asymptomatic	2 (7.5%)	3 (13%)

**Table 3 brainsci-13-00361-t003:** Literature review of viral illness associated with PTC in.

Ref.	Etiology	Number	Age	Gender	Presentation Time	Fever	CSF
[3]	No pathogen	15	6–10 months	6 M 9 F	1–3 weeks	12+	0–3 cells
[4]	Hep E	1	7 years	M	9 days	+	0
[5]	Hep A	1	4 years	M	3 days	+	0
[6]	Measles	1	8 years	F	3 weeks	-	0
[7]	varicella	1	8 years	F	3 weeks	-	Thrombosis
[8]	varicella	7	6–15 years	3M 4F	5 days–3 weeks	-	3 pleocytosis
[9]	Zoster	1	14 years	F	Immediately	-	pleocytosis
[11]	Enterovirus 71	1	6 years	F	Immediately	+	pleocytosis

## Data Availability

Data is unavailable due to privacy restrictions.

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
