# Peer review of "Prospective, Cross-Sectional Study Finds No Common Viruses in Cerebrospinal Fluid of Children with Pseudotumor Cerebri"

_brainsci, 2023, doi:10.3390/brainsci13020361_

Round 1
Reviewer 1 Report
The rationale for the study is very interesting. The prospective nature of the study is an advantage.
Many of the limitations of the study derive from the poorly described connection between CSF infection and PCT in the literature/included papers.
The paper is concise overall but the findings of the previous cases reports/series are not clearly presented/discussed by the authors.
The authors need to organize/clarify specific aspects of the literature. A mini tabular review of the cases published so far seems essential.
Additional suggested changes/additions
- The institutions cited as associated with the authors seem unrelated to the 3 centers in Israel described in the abstract/manuscript - please correct
- The absence of pleocytosis in CSF despite the presence of viral infection should be emphasized in the introduction to highlight the rationale for the study
- Hepatitis A and enterovirus have been associated with PTC according to the authors. However, these 2 viruses are not tested in the panel used. This has to be included in the limitations section of the manuscript.
- The authors should provide in a table the timing of CSF retrieval when compared to the timing of PCT diagnosis in each patient and the mean with SD.
- The authors should list/provide in their discussion timelines regarding CNS infection and PCT diagnosis gathered from the papers they cite such as Ravid et al., Konrad et al. TaÅŸdemir et al., Thapa and colleagues etc. Did the infection precede the diagnosis of PCT ? Did they coincide? How many of these cases had pleocytosis?
A tabular review including the aforementioned approximate timeline and also the presence/absence of pleocytosis in CSF would help the reader significantly. These findings from the literature need to be presented in a more systematic way.
Thanks
Author Response
Dear Editor
Thank you for your comments and thank you for giving me the opportunity to improve my manuscript.
Enclosed my response to the reviewer comments:
The rationale for the study is very interesting. The prospective nature of the study is an advantage.
Many of the limitations of the study derive from the poorly described connection between CSF infection and PCT in the literature/included papers.
The paper is concise overall but the findings of the previous cases reports/series are not clearly presented/discussed by the authors.
Response Thank you for your comment. We added a table (3) and made changes in the discussion. WE hope that now the previous results are more clear to the reader.
The authors need to organize/clarify specific aspects of the literature. A mini tabular review of the cases published so far seems essential.
Response Thank you for your comment. We added a table (3)
Additional suggested changes/additions
The institutions cited as associated with the authors seem unrelated to the 3 centers in Israel described in the abstract/manuscript - please correct
Response – Thank you for your comment, the mistake was done by the publisher the correct affiliation is
a Department of Pediatric Neurology, Schneider Children's Medical Center of Israel, Petah Tikva, 4920235 Israel
b Sackler Faculty of Medicine, Tel Aviv University, Tel Aviv, Israel
c Pediatric Neurology Unit, Hillel Yaffe Medical Center, Hadera Israel
d Bruce Rappaport Faulty of Medicine, Technion, Haifa, Israel
e Microbiology Lab, Bnai Zion Medical Center, Haifa 3104802 Israel
f Pediatric Department Bnai Zion Medical Center, Haifa 3104802 Israel
The absence of pleocytosis in CSF despite the presence of viral infection should be emphasized in the introduction to highlight the rationale for the study
Response Thank you for your comment, the remark was added to the introduction.
Hepatitis A and enterovirus have been associated with PTC according to the authors. However, these 2 viruses are not tested in the panel used. This has to be included in the limitations section of the manuscript.
Response Thank you for your comment, we added the information to the limitation section.
The authors should provide in a table the timing of CSF retrieval when compared to the timing of PCT diagnosis in each patient and the mean with SD.
Response Thank you for your comment. The CSF was taken as part of the PTC diagnosis at the time of measuring the opening pressure. The information was added to the manuscript.
The authors should list/provide in their discussion timelines regarding CNS infection and PTC diagnosis gathered from the papers they cite such as Ravid et al., Konrad et al. TaÅŸdemir et al., Thapa and colleagues etc. Did the infection precede the diagnosis of PTC ? Did they coincide? How many of these cases had pleocytosis? A tabular review including the aforementioned approximate timeline and also the presence/absence of pleocytosis in CSF would help the reader significantly. These findings from the literature need to be presented in a more systematic way.
Response Thank you for your comment. We added a table (3) including time of presentation and CNS findings.
I hope my clarifications and revisions will make my paper suitable for publication in our journal.
Sincerely
Jacob Genizi

Reviewer 2 Report
The present study deals with the possible association between CNS infections and PTCS in children.
Although the overall number of samples could be larger, the study is presented in a decent and clear way.
This reviewer approves the publication of the study. Further research is needed to unfold the possible pathophysiological paths that connect CNS infetions and PTC syndrome.
Author Response
Thank you very much for your praising review.
Reviewer 3 Report
In this study the authors aimed to characterize PTC in children and investigate a possible role of viral infection of the central nervous system in its pathogenesis. They found that PTC was not found to be associated with the presence of viruses in CSF. Some concerns and suggestions are listed as below:
In lines 4-5, a-f was used in the author list, while 1-5 was used in lines 6-14. Please double check.
The authors said that this study was conducted in three centers in Israel. However, all authors come from Italy. Please clarify.
In this study real-time polymerase chain reaction (PCR) was used to test for the presence of 12 common viruses. However, the title mentioned 'no viruses'. How about other rare viruses? This point should be discussed.
I have read the whole manuscript and this study may not be a prospective study. Please clarify.
How many individuals were vaccinated? This may have effects on final results.
How did you obtain CSF from the control group in this study?
The authors said that all participants provided written informed consent. However, all of them are children (Participants were 50 children aged 0.5–18 years).
How many individuals were excluded in this study? How the participants were selected? Consecutive enrollment?
In the result part, the authors mentioned table 1 and table 2. However, I did not find them in the main text.
In the result part, please provide statistical analysis (for example, p value).
Some may suggest that Next Generation Sequencing should be used. How about the sensitivity of PCR for detecting 12 common viruses?
Apart from CSF, blood samples should also be tested.
In line 89, 'pseudotumor cerebri' should be 'PTC'.
The direct evidence is lacking in this study. This is a major concern.
Different regions (or populations) may have distinct virus infection status. This point should be discussed.
What about previous infection in these individuals in this study?
In the part of discussion, authors' own results should also be discussed.
Regarding CSF, did the authors perform the test immediately? Or the samples were stored before the test?
In line 136, 'large' is not appropriate.
Regarding PTC and viral pathogen, cellular and molecular mechanisms should be discussed.
Some may argue that the PCR method has limations in this study.
The presence of viruses should be tested at different disease stages.
Follow-up data should be provided.
Author Response
Dear Editor
Thank you for your comments and thank you for giving me the opportunity to improve my manuscript.
Enclosed my response to the reviewer comments:
In this study the authors aimed to characterize PTC in children and investigate a possible role of viral infection of the central nervous system in its pathogenesis. They found that PTC was not found to be associated with the presence of viruses in CSF. Some concerns and suggestions are listed as below:
In lines 4-5, a-f was used in the author list, while 1-5 was used in lines 6-14. Please double check.
Response Thank you for your comment, the mistake was done by the publisher please see below the correct affiliations.
The authors said that this study was conducted in three centers in Israel. However, all authors come from Italy. Please clarify.
Response – Thank you for your comment, the mistake was done by the publisher the correct affiliations are:
a Department of Pediatric Neurology, Schneider Children's Medical Center of Israel, Petah Tikva, 4920235 Israel
b Sackler Faculty of Medicine, Tel Aviv University, Tel Aviv, Israel
c Pediatric Neurology Unit, Hillel Yaffe Medical Center, Hadera Israel
d Bruce Rappaport Faulty of Medicine, Technion, Haifa, Israel
e Microbiology Lab, Bnai Zion Medical Center, Haifa 3104802 Israel
f Pediatric Department Bnai Zion Medical Center, Haifa 3104802 Israel
In this study real-time polymerase chain reaction (PCR) was used to test for the presence of 12 common viruses. However, the title mentioned 'no viruses'. How about other rare viruses? This point should be discussed.
Response Thank you for your comment the title was changed to:
Prospective, Cross-sectional Study Finds No Common Viruses in Cerebrospinal Fluid of Children with Pseudotumor Cerebri
I have read the whole manuscript and this study may not be a prospective study. Please clarify
Response The CSF samples and the clinical data was studied prospectively, a clarification was added to the manuscript.
How many individuals were vaccinated? This may have effects on final results.
Response All precipitants were vaccinated for varicella-zoster virus, but not against any other pathogen included in the study. The information was added to the manuscript.
How did you obtain CSF from the control group in this study?
Response The control group were children who underwent lumbar puncture for medical reasons but their opening pressure was normal.
The authors said that all participants provided written informed consent. However, all of them are children (Participants were 50 children aged 0.5–18 years).
Response Thank you for your comment, the informed consent was obtained by the children care givers, clarification was added to the manuscript.
How many individuals were excluded in this study? How the participants were selected? Consecutive enrollment?
Response The participants were enrolled consecutively, non was excluded.
In the result part, the authors mentioned table 1 and table 2. However, I did not find them in the main text.
Response The tables were added as supplementary material and are enclosed:
p |
Age (years) |
p |
Total |
Boys |
Girls |
|
P=0.29 |
12.1 (4.3–17.7) |
P=0.35 |
27 |
14 (52%) |
13 (48%) |
PTC |
10.7 (0.05–17.4) |
23 |
16 (70%) |
7 (30%) |
Control group |
Table 1: Characteristics of the PTC and control groups.
Table 2: Clinical presenting symptoms of the PTC and control groups.
Control |
PTC |
|
8 (35%) |
19 (70%) |
Headache |
4 (17%) |
6 (22%) |
Vomiting |
7 (30%) |
14 (52%) |
Visual changes |
0 |
6 (22%) |
Tinnitus |
(13%) 3 |
(7.5%) 2 |
Asymptomatic |
In the result part, please provide statistical analysis (for example, p value).
Response Thank you for your comment, p value was added to table 1.
Some may suggest that Next Generation Sequencing should be used. How about the sensitivity of PCR for detecting 12 common viruses?
Response The sensitivity of the panel was 100 copies/reaction.
Apart from CSF, blood samples should also be tested.
Response Thank you for your comment the suggestion was added to the limitation section.
In line 89, 'pseudotumor cerebri' should be 'PTC'.
Response Thank you for your comment the text was changed accordingly.
Different regions (or populations) may have distinct virus infection status. This point should be discussed.
Response Thank you for your comment we added the remark in the discussion.
What about previous infection in these individuals in this study?
Response Thank you for your comment. The purpose of our study was to look for an acute infection and not to evaluate post infections syndromes and because of that PCR was chosen. However the remark was added to the limitation section.
In the part of discussion, authors' own results should also be discussed.
Response Thank you for your comment the discussion was edited accordingly.
Regarding CSF, did the authors perform the test immediately? Or the samples were stored before the test?
Response The samples were kept in -80 Celsius and the PCR panel was performed at the end of the study. The information was added to the manuscript.
In line 136, 'large' is not appropriate.
Response Thank you for your comment the text was changed to: Our study systematically examined the cerebrospinal fluid of a group of children with PTC
Regarding PTC and viral pathogen, cellular and molecular mechanisms should be discussed.
Response Thank you for your comment, a paragraph discussing the cellular and molecular mechanisms was added to the discussion.
Some may argue that the PCR method has limations in this study. The presence of viruses should be tested at different disease stages.
Response Thank you for your comment. Indeed, although PCR of the spinal fluid is a good method to evaluate an acute infection, it is not a good method to evaluate post infections syndromes. The purpose of our study was to look for an acute infection and because of that PCR was chosen. However the remark was added to the limitation section.
Follow-up data should be provided.
Response The follow up study is still ongoing and the data is not available.
I hope my clarifications and revisions will make my paper suitable for publication in our journal.
Sincerely
Jacob Genizi

Round 2
Reviewer 3 Report
The authors have addressed my concerns.